# Who has never tested for HIV following a community-based distribution of HIV self-test kits? Establishing associated predictors in rural Zimbabwe

Wellington Murenjekwa[1,2*], Kudzai Chidhanguro[1], Frances M. Cowan[1,3], Fiona C. Lampe[4], Cheryl Johnson[5], Amon Mpofu[6], Getrude Ncube[2], Owen Mugurungi[2], Karin Hatzold[7], Elizabeth L. Corbett[8,9], Andrew N. Phillips[4], Euphemia Sibanda[1,3], Valentina Cambiano[4]

**1** CeSHHAR Zimbabwe, Harare, Zimbabwe, **2** AIDS and TB Unit, Ministry of Health and Child Care, Harare, Zimbabwe, **3** Department of International Public Health, Liverpool School of Tropical Medicine, Liverpool, United Kingdom, **4** Institute for Global Health, University College London, London, United Kingdom, **5** Global HIV, Hepatitis and STI Programmes, World Health Organisation, Geneva, Switzerland, **6** National AIDS Council, Harare, Zimbabwe, **7** Population Services International, Cape Town, South Africa, **8** Department of Clinical Research, London School of Hygiene & Tropical Medicine, London, United Kingdom, **9** TB-HIV Group, Malawi-Liverpool-Wellcome Trust Clinical Research Programme, Blantyre, Malawi

* wellington.murenjekwa@ceshhar.org

## Abstract

In 2023, Zimbabwe attained the 95-95-95 UNAIDS targets. However, some sub-populations are substantially less likely to have tested for HIV. Knowledge of characteristics of these groups is crucial in designing interventions that address their needs. We estimated the prevalence and predictors of "never-having tested for HIV" status following community-based distribution of HIV self-test kits in rural Zimbabwe. We analysed data from a household survey conducted as part of a cluster randomised trial comparing two community-based HIVST distribution models in six rural districts in 2018-19. HIVST distribution was conducted over one month, followed by the household survey after four months. Survey participants aged 16 years and above completed self-administered Audio-Computer-Assisted-Survey-Instrument. Unadjusted and adjusted mixed effect logistic regression was used to identify factors associated with never-having-tested for HIV. Of the 11,076 analysed participants, the median (IQR) age was 32(22,45) years and 54.5% were female. Seventeen percent of participants had never tested for HIV, primarily due to a perceived lack of HIV risk (50%). Never testers were more likely to be: men (adjusted odds ratio [AOR]=1.69;95%Confidence Interval [CI]=1.52–1.87); younger (16-24 years (AOR=3.84; 95%CI=3.23-4.55), 25-34 years (AOR=1.30; 95%CI=1.07–1.59)) and at-least 45 years old: (AOR=2.17; 95%CI=1.80-2.60); having lower levels of education: primary/less (AOR=1.68; 95%CI=1.46-1.98), some secondary (AOR=1.62; 95%CI=1.42-1.86) compared to at least complete secondary, unemployed (AOR=1.39; 95%CI=1.15–1.69); never married (AOR=3.48; 95%CI=2.98-4.07) and previously married (AOR=1.41; 95%CI=1.19-1.68) compared to currently married; having stigmatizing beliefs

**Data availability statement:** Data was uploaded as Supporting information.

**Funding:** This work was supported by the Medical Research Council (grant number MR/T042796/1); UNITAID STAR project (PO# 10140-0-600 and PO# 8477-0-600 to WM). The funders had no role in study design, data collection and analysis, decision to publish, or preparation of the manuscript.

**Competing interests:** The authors have declared that no competing interests exist.

(AOR=1.42; 95%CI=1.24-1.62); having: low (AOR=1.52, 95%CI=1.32-1.74) and medium (OR=1.53, 95%CI=1.33-1.75) levels of treatment optimism; not participating in household decisions (AOR=1.96; 95%CI=1.70-2.27) and not reporting condomless sex (AOR=2.58; 95%CI=2.31-2.87). The Ministry of Health need to scale up acceptable and targeted interventions to improve HIV testing in different subpopulations which includes but not limited to young people, unmarried, unemployed, those with stigmatizing beliefs and those not participating in decision making.

## Author summary

Despite the increase in number of HIV testing approaches and of HIV tests, we still have many people who have never tested for HIV in their lifetime. Knowing the characteristics of people who have never tested for HIV, even after free distribution of HIV self-testing kits in their communities, is important to develop testing strategies that address their needs. We, therefore, estimated the prevalence of never having tested for HIV in these rural communities and identified the associated characteristics. We found that about one in five had never tested for HIV and half of these reported "no HIV risk perception" as the reason for not testing. Characteristics associated with not testing were being young, having a low level of education, being unemployed, unmarried, lack of involvement in decision making, having stigmatizing beliefs, not believing that antiviral therapy (ART) can improve health outcomes among people living with HIV and having sex using condom. There is need to offer acceptable and population specific interventions, to improve HIV testing in different subpopulations.

## Introduction

HIV has claimed many lives globally and an estimated 38.4 million people were living with HIV by end of 2021 [1,2]. Globally, the burden of HIV is highest within the Eastern and Southern Africa (ESA) region which accounts for more than half (54%) of all people living with HIV (PLHIV) in the world [3]; while accounting for 7% of the world population [4].

Although the estimated numbers of new HIV infections has declined by 44% and estimated AIDS related deaths by 58% between 2010 and 2021in ESA compared with other regions [3], more needs to be done if the region is to achieve the goal of ending AIDS as a public health threat by 2030 [5]. In Zimbabwe, new HIV infections decreased by an estimated 40% between 2015 and 2021 [3] and HIV prevalence among adults 15-49 years fell from 18.1% in 2005 [6] to 11.8% in 2020 [7]. The reduction in HIV incidence and prevalence was mainly due to reduction in sexual risk behaviour [8,9], increase in the number of people living with HIV on antiretrovirals [10], and implementation of the Extended Zimbabwe National HIV and AIDS Strategic Plan (ZNASP III) 2015-2020 [11], which aimed to achieve the 90-90-90 targets by 2020 and focused on high-impact interventions for key and vulnerable populations. Key program activities included combination HIV prevention strategies, social and behaviour change communication campaigns, widespread condom promotion and distribution, the promotion of male circumcision and robust prevention of mother-to-child transmission programs.

Despite the country achieving 95-95-95 [12,13] UNAIDS 2025 targets (95% of people living with HIV knowing their status, 95% of those who know their status receiving treatment and 95% of those on treatment being virally suppressed), there are specific subpopulations, such

as men and key populations (female sex workers and men who had sex with men (MSM)) and especially young people (including children), that are still lagging behind [14–16]. These inequalities are probably due to lack of innovative case finding in children [16], offer of HIV testing being primarily in clinics and discriminating laws against MSM and sex workers [14].

HIV testing is the gateway to HIV prevention, care, and treatment services. In 2014 the Ministry of Health and Child Care (MOHCC) highlighted the need for increased access to HIV testing services, given the compelling HIV infection rates in the country [17]. In 2016 MOHCC then recommended HIV self-testing (HIVST) as an additional approach to HIV testing services [18] following extensive STAR consortium work on HIVST. Although willingness and interest to use HIVST was quite high, access to and awareness of HIVST was low in Zimbabwe [19].

The annual number of conventional HIV tests performed in Zimbabwe increased from 1.7 million in 2011 to 3 million in 2018 [20] and, as a consequence, the percentage of adults (15-49 years old) who were ever tested and had received results for HIV: among men from 16.4% in 2005 to 76.3% in 2020 and among women from 21.7% in 2005 [21] to 84.9% in 2020 [7]. This increase in number of HIV tests is the result of different testing approaches introduced by the MOHCC: provider-initiated testing and client-initiated testing, HIVST (facility- and community-based) with those with reactive results being confirmed with conventional testing, and index testing. Despite the expansion in HIV testing approaches and increase in number of tests, there is still a considerable number of people who have never tested for HIV in their lifetime. Knowledge of characteristics of people who have never tested for HIV, particularly in the context of community-based promotion of HIV testing, is crucial for developing testing models that target those specific subpopulations. Therefore, this study estimated the prevalence of never having tested for HIV among people living in rural areas of Zimbabwe and identified the associated characteristics. We also assessed factors associated with reporting no perceived HIV risk among participants who never tested for HIV.

## Methods

### Ethics statement

The research used anonymized secondary data from a study conducted by the author team, previously approved by Medical Research Council of Zimbabwe (MRCZ/A/2323), London School of Hygiene & Tropical Medicine Ethics Committee (15801) and WHO Ethical Review Committee (ERC.0003065); and informed written consent was obtained from all the participants. Prior to conducting the research, verbal consent to engage with the communities was obtained from local community leaders (headmen, village heads and ward councillors).

### Study design and settings

The study analysed data from a cluster randomised trial (CRT) comparing two community-based HIVST distribution models, conducted between 2018 and 2019, in six rural districts (Shamva, Muzarabani, Mutoko, Zvimba, Shurugwi and Umguza) in five provinces of Zimbabwe where Population Service International (PSI) was implementing HIV testing services. Detailed information of the CRT was described elsewhere [22].

Briefly, in the CRT, clusters were defined as headman units, which are administrative units in rural areas of Zimbabwe responsible for implementing community level activities. Forty clusters were randomly allocated to either community-led or paid HIVST distribution arms using 1:1 restricted randomisation. To be considered potentially eligible, clusters had to have at least three census enumeration areas (EAs) within its boundaries, not share a health facility with a neighbouring cluster and the distance between any two selected clusters was at least 20

km. Randomisation was restricted on district, distance from the centre of the headman unit to the nearest health facility and availability of PrEP at the nearest health facility. Consent for communities to participate was sought and obtained from the community leaders.

In the community-led arm, community members at least 16 years of age were invited to the first meeting during which they were introduced to the concept of a community-led model and the importance of HIV testing. The concept of antiretroviral treatment (ART) for preventing HIV transmission within their headman unit and the potential to reduce new HIV infections through early HIV testing and linkage to ART was discussed and explained. Members were entrusted with the responsibility of developing their own HIVST kits distributing models with the options of selecting distributors, deciding on incentives for distributors, managing of kits and how to ensure adherence to regulatory requirements. The distribution models had to adhere to the following MOHCC guidelines testing criteria: kits only to be given to individuals 16 years old and above (age of consent for HIV testing), no forced testing or results disclosure, and confidentiality to be upheld. HIVST distributors received a 3-day training program according to MOHCC guidelines. The training covered areas on HIV testing, facilitating utilization of HIVST kits by others, sharing information to promote and facilitates access to appropriate post-test services, providing information on effectiveness of ART for preventing HIV, and how to complete the data collection tool. Distribution of kits was done over a period of 4-6 weeks.

In the paid distributor model, distributors received the same 3-day training as in the community led model and had 4 weeks to conduct door-to-door distribution in one or two villages. PSI provided training, kits, and a one-off stipend of US$50 to distributors.

In both HIVST distribution models, distributors were expected to give kits to all willing individuals (one kit per person) in their geographical area who met the testing criteria and share information about linkage to all post-test services that testers could access from local clinics.

## Study population and data collection

Trial outcomes were evaluated using household survey 4 months post-distribution to determine uptake of HIVST and linkage to post-test services. All eligible individuals (16 years and above) residing in the selected enumeration areas during HIVST distribution, upon giving written consent to participate in the survey, completed a self-administered Audio Computer-Assisted Survey Instrument questionnaire (ACASI) between 08 October 2019 and 30 December 2019.Tablet computers and headsets were used to enhance participants comfort and ability to respond freely. In this analysis, we excluded participants who provided inconsistent answers regarding never having tested for HIV: those who reported having ever had a positive HIV result, despite reporting never having had an HIV test.

## Study variables

We defined the main outcome of never having tested for HIV as having answered "No" to the question "Have you ever been tested for HIV including HIVST?". The question was asked to all participants and the response options were: "Yes"/"No". The secondary outcome of no perceived HIV risk was defined as having selected "Not at risk of contracting HIV infection" to the question "What best describes why you haven' t tested for HIV?" and then having selected "Not at risk of contracting HIV infection" to the follow up question: "Of the factors you just mentioned, which was the most important reason you have not tested for HIV?". The selected explanatory variables includes: socio-demographic characteristics (age, sex, education, religion, marital status, employment status, steady partner status, household head status, wealth

quintile), perceived health status, participation in decisions (about their health, household expenditure and visiting relatives), community cohesion(social cohesion, critical consciousness, shared concern), stigma(perceived stigma in community, stigma-any negative attitudes, stigma in health care settings), condomless sex in the past three months and optimism on effectiveness of antiretroviral therapy (ART), referred to as "treatment optimism". (Refer to S1 Table, for a comprehensive list of explanatory variables and their levels. The next paragraph gives a summary of how certain variables were derived. Six variables had missing data, and the frequency of missing or "declined to answer" combined was low: perceived health status (1.0%), employment status (1.1%), religion (1.4%), steady partner status (1.7%), treatment optimism (5.0%) and wealth quintile (5.1%). Individuals with missing data were excluded only from the analyses in which the respective missing variable was included when investigating factors associated with never having tested for HIV (ie in the regressions).

**Construct explanatory variables.** The derived variables were: individual level variables (wealth quintile, participation in decisions, stigma (any negative attitude), and treatment optimism) and cluster level variables(community cohesion sub-scales, and perceived stigma community).

The wealth quintile variable was constructed using data on seventeen binary response questions concerning items owned by surveyed households. As done in the primary study by Sibanda E L et al [22] we ran a polychoric principal component analysis on the dummy variables as recommended and supported by the literature [23–25] and then derived household scores based on the first principal component. The scores were then grouped into quintiles of household index and participants within a household were assigned a quintile level for that household [22].

The variable "participation in decisions" was constructed using three questions asking respectively who makes decisions about: health care, major household purchases and visiting participant's relatives. Participants who made decisions either alone or jointly with partners on all three questions were coded as 2 (three decisions), those who decided alone or jointly with partners in any one or two questions were coded as 1 (one/two decisions), while those who indicated not participating in any of the decisions were coded as 0 (zero decisions).

Factor analysis, a variable reduction technique with one factor solution using the principal component factoring (PCF) method was conducted on community cohesion subscales (social cohesion, shared concern, and critical consciousness), stigma scales ("perceived stigma in community", "stigma: any negative attitude (fear and judgement)", "stigma in health care settings") and "treatment optimism". The factor solution was then rotated using the promax method.

The factor scores for community cohesion subscales and "perceived stigma in community" scale were then summarised at a cluster level using the median, and finally the clusters were divided in tertiles and each cluster labelled as either low, medium or high based on the calculated median. The "stigma: any negative attitude (fear and judgement)" and "treatment optimism" scales scores were divided into tertiles at individual level and each individual was labelled respectively as either having low, medium or high stigma and low, medium or high "treatment optimism".

A total of 27 questions were related to community cohesion (See S2 Table). These were grouped into three subscales: social cohesion (6 items), shared concern (10 items) and critical consciousness (11 items). Response options for the items were: strongly agree, somewhat agree, neither agree nor disagree, somewhat disagree, and strongly disagree. The internal consistencies of the subscales as measured by Cronbach alpha were 88% (social cohesion), 92% (shared concern) and 95% (critical consciousness). The community cohesion questions were adopted from the Lipman Study [26]. The variance explained by the first factors of the sub-construct variables was as follows: 64% for "social cohesion", 58% for "shared concern", and 66% for "critical consciousness".

Ten questions related to stigma (see S2 Table) were adapted from the Hargreaves paper and had previously been validated [27]; they were grouped into 3 subscales: "stigma in health care settings" (2 items), "perceived stigma in community" (5 items) and "stigma: any negative attitude (fear and judgement)" (3 items). Response options on a Likert scale were strongly agree, agree, unsure, disagree, and strongly disagree. Cronbach alpha for the sub-scales were 66% ("stigma in healthcare settings"), 78% ("perceived stigma in the community") and 78% (any negative attitude (fear and judgement)). The variance explained by the first factors of the sub-construct variables was, 70% for "stigma: any negative attitude (fear and judgement)" and 53% for "perceived stigma in the community".

The "treatment optimism" variable was measured through four questions obtained from WHO generic tools for operations research [28] (see S2 Table), rated on the same 5-point Likert scale as the questions related to stigma. The Cronbach alpha was 75%, with the first factor explaining 58% of the variance.

Cronbach alpha and Kaiser-Meyer Olkin (KMO) measure of sampling adequacy (MSA) values for the constructs were all within the acceptable range of 70% to 100% [29–31], except for "stigma in health care settings" where both the Cronbach alpha(66%) and the KMO MSA (50%) fell below the acceptable range. As a result, "stigma in health care settings" was excluded from the analysis.

See S2 Table; for further details on community cohesion, stigma, and "treatment optimism" scales.

**Inclusivity in global research.** Additional information regarding the ethical, cultural, and scientific considerations specific to inclusivity in global research is included in S1 Text.

## Data analysis

Categorical variables were described using frequencies and proportions. The median (IQR) was used to summarise the age of respondents. Unadjusted and adjusted mixed effects logistic regression with a random intercept at headman unit level (to account for clustering within each headman unit) was used to model the relationship between independent variables and the outcome. We first examined each factor separately and then we considered two adjustments: adjustment for sex, and age (Model I) and adjustment for sex, age, religion, and education (Model II). This strategy was done to reduce the risk of over adjustment. We presented the results from the two models to show the effect of additional adjustment for religion and education. The "perceived stigma in community" variable and community cohesion subscale variables were used as cluster level predictors in the analysis, while other variables were used as individual level predictors. We also tested for interaction (using likelihood ratio test) between sex and condomless sex in the past 3 months, to assess whether the effect of condomless sex in the past 3 months on never having tested for HIV varied between men and women.

Using the same independent variables, we also analysed factors associated with no perceived HIV risk among participants who reportedly never tested for HIV. Results are presented as unadjusted and adjusted odds ratios with 95% confidence intervals. All statistical analyses were performed using STATA version 17.

## Results

Of the 11,150 survey participants recruited to the community-led CRT, 74 (<1%) were excluded from the analysis as they provided inconsistent replies regarding the outcome of interest (never tested for HIV but reported an HIV positive result).

Table 1 presents the characteristics of study participants included in the analysis. The median (IQR) age of participants was 32 (22,45) years and 54.5% were women. Nearly half of

**Table 1. Descriptive characteristics of participants (n=11,076).**

|  | Ever tested | Never tested | Total |
|---|---|---|---|
| **All** | **[9254/11076]** **83.5%** | **[1822/11076]** **16.5%** | **11076** **(100%)** |
| **Population Characteristics** | | | |
| **Age (years),[n] median(IQR)** | [9254] 33 (24,45) | [1822] 25 (18,45) | [11076] 32 (22,45) |
| **Age category** | | | |
| 16-24 years, n (%) | 2518 (73.8%) | 894 (26.2%) | 3412 (30.8%) |
| 25-34 years, n (%) | 2357 (89.4%) | 281 (10.6%) | 2638 (23.8%) |
| 35-44 years, n (%) | 2014 (91.7%) | 183 (8.3%) | 2197 (19.8%) |
| 45 + years, n (%) | 2365 (83.6%) | 464 (16.4%) | 2,829 (25.5%) |
| **Sex** | | | |
| Male, n (%) | 4003 (79.5%) | 1034 (20.5%) | 5037 (45.5%) |
| Female, n (%) | 5251 (87.0%) | 788 (13.0%) | 6039 (54.5%) |
| **Household head status** | | | |
| Household head, n (%) | 3840 (86.5%) | 601 (13.5%) | 4441 (40.1%) |
| Household head rep, n (%) | 1062 (89.3%) | 127 (10.7%) | 1189 (10.7%) |
| Not household head/rep, n (%) | 4352 (79.9%) | 1094 (20.1%) | 5446 (49.2%) |
| **Education** | | | |
| Primary Complete/less, n (%) | 3630 (83.5%) | 718 (16.5%) | 4348 (39.3%) |
| Some secondary, n (%) | 2209 (79.7%) | 552 (20.3%) | 2794 (25.2%) |
| Secondary complete/tertiary, n (%) | 3397 (86.4%) | 537 (13.6%) | 3934 (35.5%) |
| **Employment** | | | |
| Unemployed, n (%) | 7082 (82.9%) | 1465 (17.1%) | 8547 (77.2%) |
| Self-employed/subsistence farmer, n (%) | 1031 (86.2%) | 165 (13.8%) | 1196 (10.8%) |
| Formally employed, n (%) | 1053 (87.3%) | 153(12.7%) | 1206 (10.9%) |
| Missing, n (%) | 88(69.3%) | 39 (30.7%) | 127(1.1%) |
| **Religion** | | | |
| Apostolic, n (%) | 3636 (85.9%) | 599 (14.1%) | 4235 (38.2%) |
| Catholic/ Protestant, n (%) | 2071 (81.9%) | 458 (18.1%) | 2529 (22.8%) |
| Pentecostal, n (%) | 1145 (87.1%) | 169 (12.9%) | 1314 (11.9%) |
| No religion/ATR, n (%) | 1421 (79.5%) | 367 (20.5%) | 1788 (16.1%) |
| Other, n (%) | 981 (81.1%) | 229 (18.9%) | 1210 (10.9%) |

*(Continued)*

**Table 1.** (Continued)

| | Ever tested | Never tested | Total |
|---|---|---|---|
| **Marital status** | | | |
| First marriage (currently), n (%) | 3636 (89.5%) | 599 (10.5%) | 5272 (47.6%) |
| Remarried after divorce/widowed, n (%) | 2071 (90.9%) | 458 (9.1%) | 1449 (13.1%) |
| Previously married (Widowed/Separated/Divorced), n (%) | 1145 (85.2%) | 169 (14.8%) | 1626 (14.9%) |
| Never married, n (%) | 1421 (67.3%) | 367 (32.7%) | 2576 (23.3%) |
| Missing, n(%) | 101 (66.0%) | 52 (33.0%) | 153 (1.4%) |
| **Current steady partner** | | | |
| No, n (%) | 2,792(72.9%) | 1,039 (27.1%) | 3831 (34.6%) |
| Yes, n (%) | 6329 (89.7%) | 726 (10.3%) | 7055 (63.7%) |
| Missing, n(%) | 133 (70.0%) | 57 (30.0%) | 190 (1.7%) |
| **Health status:** | | | |
| Very good, n (%) | 2606 (81.8%) | 578 (18.2%) | 3184 (28.8%) |
| Good, n (%) | 2959 (82.4%) | 631 (17.6%) | 3590 (32.4%) |
| Fair, n (%) | 2493 (85.4%) | 425 (14.6%) | 2918 (26.4%) |
| Poor, n (%) | 1118 (87.8%) | 156 (12.2%) | 1274 (11.5%) |
| Missing, n(%) | 72 (70.9%) | 32 (29.1%) | 110 (1.0%) |
| **Wealth quintile:** | | | |
| Lowest, n (%) | 1572 (83.0%) | 321 (17.0%) | 1893 (17.1%) |
| Second, n (%) | 1739 (85.9%) | 285 (14.1%) | 2024 (18.3%) |
| Middle, n (%) | 1736 (83.0%) | 354 (16.9%) | 2090 (18.9%) |
| Fourth, n (%) | 1815 (84.0%) | 345 (16.0%) | 2160 (19.5%) |
| Highest, n (%) | 1937 (82.7%) | 405 (17.3%) | 2342 (21.1%) |
| Missing, n(%) | 455 (80.3%) | 112 (19.7%) | 567 (5.1%) |
| **Number of decisions participated in (household purchases, health care and family visits)** | | | |
| Three, n (%) | 5321 (86.8%) | 807(13.2%) | 6125 (55.3%) |
| One/two, n (%) | 2802(84.9%) | 497(15.1%) | 3299 (29.8%) |
| None, n (%) | 1131(68.6%) | 518 (31.4%) | 1649 (14.9%) |
| **Condomless sex in the past 3 months** | | | |
| Yes, n (%) | 5861 (89.9%) | 655(10.1%) | 6516 (58.8%) |
| No, n (%) | 3393 (74.4%) | 1167(25.6%) | 4560 (41.2%) |
| **Social cohesion***: | | | |

*(Continued)*

**Table 1.** (Continued)

| | Ever tested | Never tested | Total |
|---|---|---|---|
| Low, n (%) | 2492(85.8%) | 413(14.2%) | 2905 (26.2%) |
| Medium, n (%) | 3037(82.1%) | 663(17.9%) | 3700 (33.4%) |
| High, n (%) | 3725(83.3%) | 746(16.7%) | 4471 (40.4%) |
| **Critical consciousness**[*]: | | | |
| Low, n (%) | 2970(83.3%) | 596(16.7%) | 3566 (32.2%) |
| Medium, n (%) | 3142(83.8%) | 609(16.2%) | 3751 (33.9%) |
| High, n (%) | 3142 (83.6%) | 617(16.4%) | 3759 (33.9%) |
| **Shared concern**[*]: | | | |
| Low, n (%) | 3042 (83.3%) | 612 (16.7%) | 3654 (33.0%) |
| Medium, n (%) | 3089 (83.2%) | 622 (16.8%) | 3711 (33.5%) |
| High, n (%) | 3123 (84.2%) | 588 (15.8%) | 3711 (33.5%) |
| **Perceived stigma in community**[*]: | | | |
| Low, n (%) | 1409(84.4%) | 261(15.6%) | 1670 (15.1%) |
| Medium, n (%) | 4683(83.9%) | 898(16.1%) | 5581(50.4%) |
| High, n (%) | 3162(82.7%) | 663(17.3%) | 3825 (34.5%) |
| **Stigma: Any negative attitude** | | | |
| Low, n (%) | 2689(86.2%) | 430(13.8%) | 3119 (28.2%) |
| Medium, n (%) | 3517(85.4%) | 601(14.6%) | 4118(37.2%) |
| High, n (%) | 3048(79.4%) | 791(20.6%) | 3839 (34.7%) |
| **Treatment optimism:** | | | |
| Low, n (%) | 2880 (82.1%) | 627 (17.9%) | 3507 (31.6%) |
| Medium, n (%) | 2882 (82.2%) | 626 (17.8%) | 3508 (31.7%) |
| High, n (%) | 3080 (87.8%) | 428 (12.2%) | 3508 (31.7%) |
| Missing, n(%) | 412 (74.5%) | 141 (25.5%) | 553 (5.0%) |

[*]Cluster level variables.

the participants (50.8%) were either household heads or household head representatives, with 40.1% being household heads. Three fifths (60.7%) had at least some secondary education level and 39.3% had at-most primary education level. More than three quarters (78.1%) were unemployed, whilst 11.1% were formally employed and 10.9% were self-employed. In terms of religion, 38.2% were of the apostolic faith, about a quarter (22.8%) were either Catholics or Protestants,16.1% were either of African tradition religion (ATR) or not religious, whilst Pentecostals and other religions constituted just over 10.0% each. In terms of marital status: almost

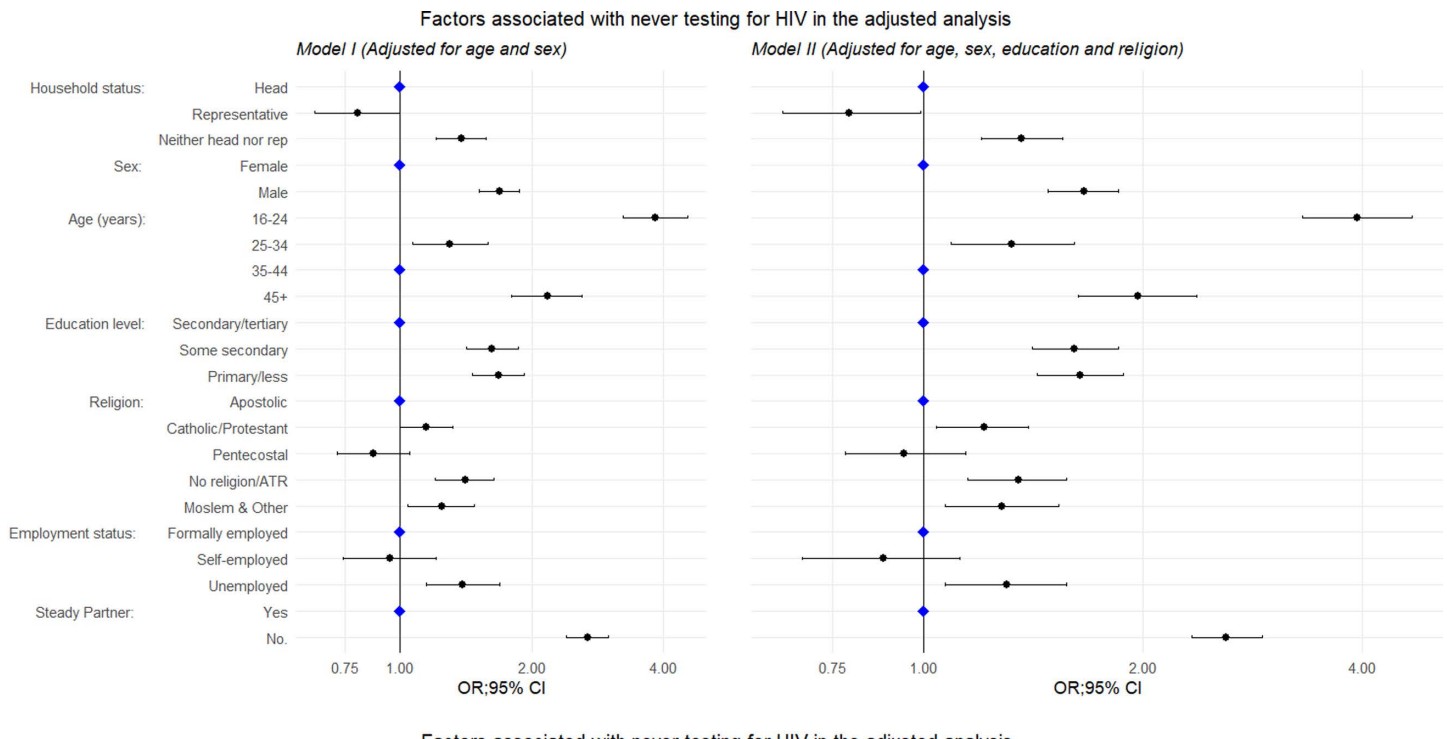

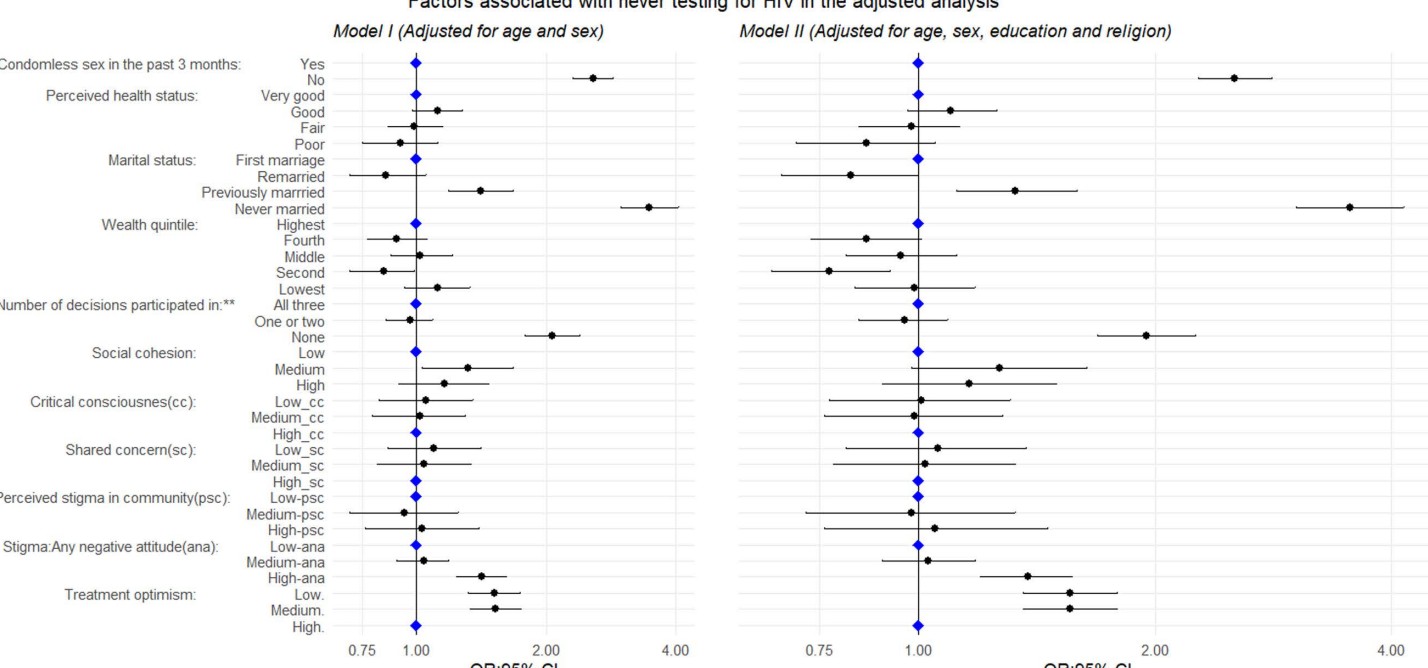

**Fig 1. Adjusted analysis of risk factors associated with never having tested for HIV (*Model I and Model II*).**

half of the participants (48.3%) were currently married in first marriage, a quarter were never married (23.6%), 13.3% had remarried after divorce or death of a partner and the remaining 14.9% were previously married (widowed/divorced). Overall, two thirds (64.8%) had a steady partner and about six in ten reported at least good health status (61.7%). Seventeen percent (1,822/ 11,076) of participants reported they had never had an HIV test.

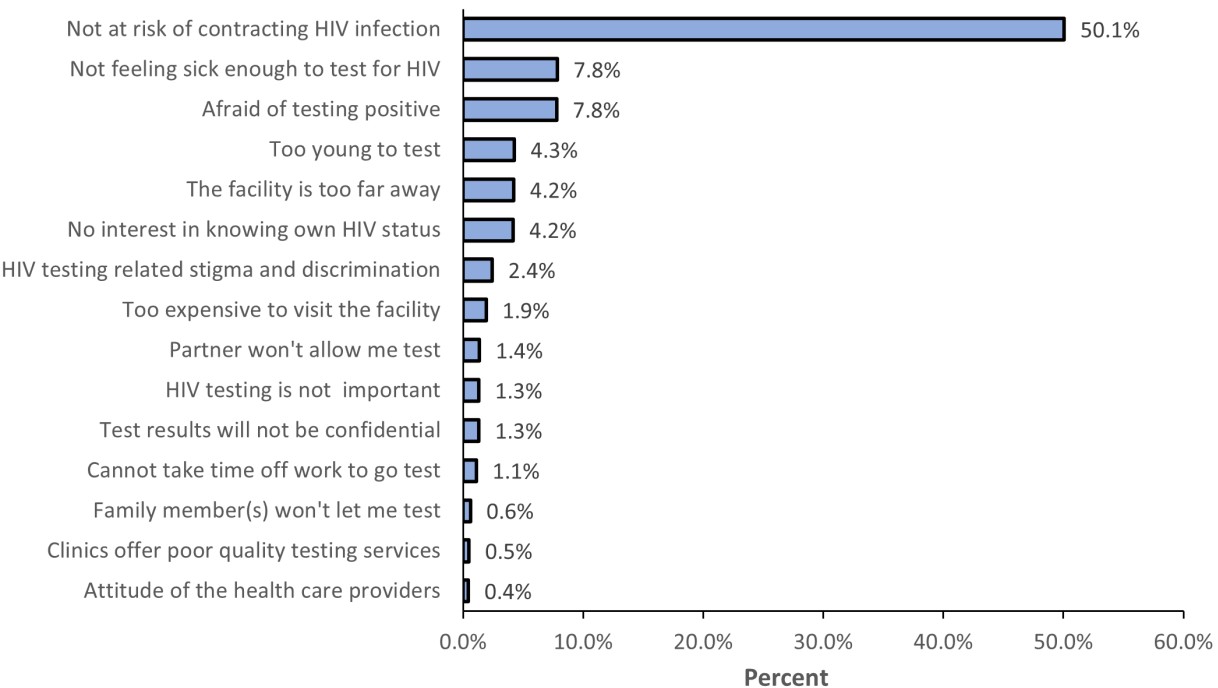

**Fig 2. Most important reason for not testing for HIV.**

Fig 1 and S3 Table present findings from the unadjusted and adjusted analysis. In the unadjusted analysis, all variables except "critical consciousness", "shared concern" and "perceived stigma in community" were significantly associated with the odds of never having tested for HIV (p<0.05). In Model I (adjusted for sex and age), being male (adjusted odds ratio [AOR]=1.69; 95% Confidence Interval [CI]=1.52–1.87); younger or older than 35-44 years: 16-24 years (AOR=3.84; 95% CI=3.23-4.55), 25-34 years (AOR=1.30; 95% CI=1.07–1.59), and at least 45 years (AOR=2.17; 95% CI=1.80-2.60), compared to the middle age group 35-44 years; having lower levels of education: at most primary (AOR=1.68; 95% CI=1.46-1.98) and some secondary (AOR=1.62; 95% CI=1.42-1.86), compared to have at least completed secondary; being unemployed (AOR=1.39; 95% CI=1.15–1.69), compared to the formally employed; being never married (AOR=3.48; 95% CI=2.98-4.07) or widowed/separated/ divorced (AOR=1.41; 95% CI=1.19-1.68), compared to being currently in the 1st marriage; with high level of HIV stigma (any negative attitudes) (AOR=1.42; 95% CI=1.24-1.62), compared to low level; with low or medium levels of treatment optimism on effectiveness of ART (respectively AOR=1.52,95%CI=1.32-1.74 and AOR=1.53,95%CI=1.33-1.75), compared to high level; being neither household-head nor household-head representative (OR=1.38,95%CI=1.21-1.57), compared to household heads; having a lower participation in any of the three areas of household decisions (AOR=2.07; 95% CI=1.79-2.39), compared to participation in all three areas; reporting no condomless sex in the past 3 months (AOR=2.58; 95% CI=231-2.87) were more likely to be never testers.

When testing the interaction between sex and condomless sex in the past 3 months, we found that the association between reporting no condomless sex and never testing was greater for women compared to men: AOR=3.53 (95% CI=2.99-4.16) vs AOR=1.99 (95% CI=1.71-2.30; p value for interaction<0.001). When additionally adjusting all associations for education

**Table 2. Factors associated with no perception of HIV risk among participants who reportedly never tested for HIV.**

| | Unad-justed OR | 95% CI | Overall P value | Adjusted OR[a] | 95% CI | Overall P value |
|---|---|---|---|---|---|---|
| **Population Characteristics** | | | | | | |
| **Age category** | | | 0.024 | | | <0.001 |
| 16-24 years | 1.60 | [1.15, 2.23] | | 1.59 | [1.14, 2.22] | |
| 25-34 years | 1.00 | [0.68, 1.47] | | 0.99 | [0.67, 1.46] | |
| 35-44 years | 1.0 | Ref | | 1.0 | Ref | |
| 45 + years | 2.30 | [1.61, 3.28] | | 2.26 | [1.58, 3.24] | |
| **Sex** | | | 0.131 | | | 0.495 |
| Female | 1.0 | Ref | | 1.0 | Ref | |
| Male | 0.86 | [0.72, 1.04] | | 0.93 | [0.77, 1.13] | |
| **Household head status:** | | | 0.519 | | | 0.498 |
| Household head | 1.0 | Ref | | 1.0 | Ref | |
| Household head rep | 1.13 | [0.76, 1.66] | | 1.19 | [0.79, 1.79] | |
| Not household head/rep | 1.10 | [0.90, 1.34] | | 1.15 | [0.90, 1.46] | |
| **Education level:** | | | <0.001 | | | 0.006 |
| Secondary complete/tertiary | 1.0 | Ref | | 1.0 | Ref | |
| Some secondary | 1.13 | [0.88, 1.44] | | 1.06 | [0.83, 1.35] | |
| Primary complete/less | 1.63 | [1.30, 2.06] | | 1.46 | [1.14, 1.87] | |
| **Employment status:** | | | 0.015 | | | 0.193 |
| Formally employed | 1.0 | Ref | | 1.0 | Ref | |
| Self employed/subsistence farmer | 1.32 | [0.84, 2.07] | | 1.23 | [0.77, 1.94] | |
| Not employed | 1.52 | [1.07, 2.15] | | 1.37 | [0.96, 1.97] | |
| **Religion:** | | | 0.033 | | | 0.255 |
| Apostolic | 1.0 | Ref | | 1.0 | Ref | |
| Catholic/ Protestant | 1.15 | [0.90, 1.48] | | 1.13 | [0.88 1.46] | |
| Pentecostal | 1.06 | [0.75, 1.50] | | 1.10 | [0.77, 1.57] | |
| No religion/ ATR | 1.15 | [0.88, 1.50] | | 1.19 | [0.89, 1.56] | |
| Moslem & Other | 1.49 | [1.09, 2.04] | | 1.44 | [1.05, 1.97] | |
| **Marital Status:** | | | 0.112 | | | 0.237 |
| 1st marriage/staying as married | 1.0 | Ref | | 1.0 | Ref | |
| Remarried after divorce/widowed | 1.00 | [0.68,1.47] | | 1.11 | [0.74, 1.64] | |
| Widowed/Separated/Divorced | 1.28 | [0.94, 1.74] | | 0.99 | [0.72,1.38] | |
| Never married | 1.18 | [0.95, 1.46] | | 1.37 | [1.01, 1.85] | |
| **Current steady partner** | | | <0.001 | | | 0.003 |
| Yes | 1.0 | Ref | | 1.0 | Ref | |
| No | 1.42 | [1.17, 1.72] | | 1.37 | [1.12, 1.69] | |
| **Perceived health status:** | | | <0.001 | | | <0.001 |
| Very good | 1.47 | [1.02, 2.10] | | 1.78 | [1.21, 2.63] | |
| Good | 1.07 | [0.75, 1.53] | | 1.24 | [0.85, 1.79] | |
| Fair | 0.95 | [0.66,1.38] | | 1.03 | [0.70, 1.51] | |
| Poor | 1.0 | Ref | | 1.0 | Ref | |
| **Household asset quintile:** | | | 0.115 | | | 0.318 |
| Highest | 1.0 | Ref | | 1.0 | Ref | |
| Lowest | 0.79 | [0.59, 1.08] | | 0.75 | [0.55, 1.02] | |
| Second | 0.86 | [0.63, 1.18] | | 0.85 | [0.62, 1.17] | |
| Middle | 0.78 | [0.58, 1.05] | | 0.77 | [0.57, 1.03] | |
| Fourth | 0.93 | [0.69, 1.25] | | 0.89 | [0.66, 1.20] | |

*(Continued)*

Table 2. (Continued)

| | Unad-justed OR | 95% CI | Overall P value | Adjusted ORᵃ | 95% CI | Overall P value |
|---|---|---|---|---|---|---|
| **Number of decisions participated in (Purchases, health care and family visits)** | | | 0.439 | | | 0.751 |
| All three | 1.0 | Ref | | 1.0 | Ref | |
| One or two | 0.95 | [0.75, 1.19] | | 0.98 | [0.76, 1.25] | |
| None | 1.11 | [0.88, 1.39] | | 1.07 | [0.83, 1.40] | |
| **Condomless sex in the past 3 months** | | | <0.001 | | | 0.002 |
| Yes | 1.0 | Ref | | 1.0 | Ref | |
| No | 1.44 | [1.18, 1.75] | | 1.40 | [1.13, 1.72] | |
| **Social cohesion:** | | | 0.724 | | | 0.331 |
| Low | 1.0 | Ref | | 1.0 | Ref | |
| Medium | 1.27 | [0.92, 1.74] | | 1.25 | [0.89, 1.73] | |
| High | 1.09 | [0.80, 1.50] | | 1.03 | [0.75, 1.42] | |
| **Critical consciousness:** | | | 0.340 | | | 0.357 |
| Low | 1.17 | [0.85, 1.60] | | 1.27 | [0.92, 1.75] | |
| Medium | 1.02 | [0.74, 1.38] | | 1.14 | [0.83, 1.56] | |
| High | 1.0 | Ref | | 1.0 | Ref | |
| **Shared concern:** | | | 0.159 | | | 0.159 |
| Low | 1.25 | [0.91, 1.70] | | 1.36 | [0.99, 1.86] | |
| Medium | 1.07 | [0.78, 1.46] | | 1.11 | [0.81, 1.52] | |
| High | 1.0 | Ref | | 1.0 | Ref | |
| **Perceived stigma in community:** | | | 0.014 | | | 0.051 |
| Low | 1.0 | Ref | | 1.0 | Ref | |
| Medium | 1.14 | [0.80, 1.62] | | 1.13 | [0.78, 1.62] | |
| High | 1.51 | [1.04, 2.20] | | 1.51 | [1.02, 2.22] | |
| **Stigma: Any negative attitude** | | | 0.231 | | | 0.069 |
| Low | 1.0 | Ref | | 1.0 | Ref | |
| Medium | 0.87 | [0.68, 1.12] | | 0.89 | [0.69, 1.15] | |
| High | 1.11 | [0.87, 1.41] | | 1.15 | [0.90, 1.47] | |
| **Treatment optimism:** | | | 0.458 | | | 0.198 |
| Low | 0.89 | [0.69, 1.14] | | 0.87 | [0.61, 1.10] | |
| Medium | 0.79 | [0.61, 1.01] | | 0.79 | [0.64, 1.07] | |
| High | 1.0 | Ref | | 1.0 | Ref | |

a-Adjusted for age and sex, CI- confidence interval; OR-Odds ratio.

and religion (Model II), the results did not change in any meaningful way, see Fig 1 and/or S3 Table.

Among participants who reported never having tested for HIV (n=1,822), the most important cited reason for never testing was no perceived HIV risk (50%) (of which a third reportedly engaged in condomless sex), followed by being afraid of testing HIV positive and not feeling sick enough to test for HIV, both 8% each as shown in Fig 2.

Factors associated with increased odds of reporting no HIV risk perception in the adjusted analysis (adjusted for sex and age) were low education levels; never married, compared to being currently in the 1st marriage; younger or older age: 16-24 years, and at-least 45 years; not having a steady partner and not reporting condomless sex, compared to those who reportedly had. Also, reporting very good health status was associated with increased odds of reporting no HIV risk perception compared with poor health, see Table 2.

## Discussion

We found that 17% of the participants had never had an HIV test, in rural communities in Zimbabwe between 2018-2019 despite a community-based distribution of HIV self-testing kits, which greatly improved access to HIV testing [32,33]. This study did not collect baseline information on HIV testing uptake in the communities and did not evaluate whether the introduction of community-based HIV self-testing increases testing rates. However, previous evidence had reported that HIVST increases uptake of HIV testing [32,33]. It is also important to provide a range of HIVST delivery models, as testing rates have been shown to improve with a variety of HIVST models and support tools [34]. These results are comparable with the findings from a household survey done in Zimbabwe in 2020 [7] which reported that 20% and 16% in rural and urban areas respectively had never tested for HIV. More needs to be done to identify non HIV testing sub-populations if the country is to meet the UNAIDS 2030 target of ending AIDS as a public health threat [5].

Never testers were more likely to be men, and this is in line with several other studies and reports [35–39]. Possible reasons are that men tend to have poorer health seeking behaviour [40,41], use partner's HIV status as substitute for testing, and that women access HIV information and testing through antenatal clinics [42]. High uptake among women has also been attributed to the integration of HIV testing into clinical services for women [39]. In this study, HIV self-testing, which was found to be highly acceptable among men [43], was widely available so our findings need to be interpreted in that context.

Our study supports findings from several studies which reported an association between age and never having tested for HIV, with younger people having higher odds of being never testers for HIV [35,36,44], however, they may be less likely to be sexually active. Young people face barriers to HIV testing and these include low HIV risk perception [45], scarce opportunities for accessing HIV testing services, not trusting in HIV testing services, and unfriendly attitudes by health care workers [46]. Among never testers, the most cited reason was no perceived HIV risk: young people (15-24 years) and older people (45+ years) were more likely to report no HIV risk perception when compared to the 35-44 years old. Considering that HIV self-test kits were distributed in the communities four weeks prior to the survey, there were plenty opportunities for accessing HIV services. Other possible explanations are fear of stigma and discrimination related to HIV testing and fear of a positive HIV result [47]. We also found that individuals aged 45 years and above, had higher odds of being never testers compared to 35-44 years age group. This is likely attributed to low HIV risk perception among the older age group as reported in this study. Studies among young people have reported a positive correlation between HIV testing uptake and risk perception [48,49]. However, the reason why those at least 45 years were unlikely to test for HIV needs to be investigated further.

In this study, lower levels of education were associated with never having tested for HIV, and this finding also emerged in studies conducted in Ethiopia [50] on HIV testing practice among women 15-24 years, in Zimbabwe [51] and several other studies [35,36,44]. Greater knowledge of HIV allows for better understanding of the benefits of testing for HIV [52] and therefore greater uptake of HIV testing [53].

Consistent with other studies [54,55], the unemployed were more likely to be never testers compared to the formally employed. Research suggests that employment is associated with increased social connections, financial freedom, and independence [56]. However, as part of this trial HIV self-testing kits were distributed for free, with instructions and demonstrations on how to use the kits. Post-test services information was shared, and recipients would test at their own convenient time without disclosing results to distributors. Therefore, further investigations are needed to understand the reasons why the unemployed were less likely to test.

Never married participants were more likely to be never testers than those who were married, even after adjusting for age. This result supports finding from previous studies in Zimbabwe, Kenya, and Ethiopia [36,44,51]. Similarly, those who were currently not in a partnership but who were previously (widowed/ separated/ divorced) were more likely to be never testers than married participants. A possible explanation is that participants who are currently not in a partnership, like those never married, are at low risk of contracting HIV [48,49].

Stigma (any negative attitudes) was associated with being a never tester. In line with our findings, previous studies have reported lower HIV testing uptake among participants with high levels of stigma [36,57,58]. Interventions designed to reduce individual stigma may help in improving uptake of HIV testing. Owing to the complexity of HIV/AIDS related stigma, efforts to reduce this stigma must address the different aspects of it which varies by culture and on whether its intrapersonal or societal [59]. Offering HIV testing as a package together with other health services such as routine medical check-ups can contribute to reduce stigma associated with HIV [60]. In addition, other ways to reduce HIV related stigma and discrimination include, but is not limited to, strengthening capacity of community health workers by ensuring appropriate linkages between communities and formal health systems; ensuring that young people have access to youth-friendly HIV services and comprehensive sexuality education; engaging communities in stigma- and discrimination-reduction activities; educating workplace communities on HIV, comorbidities and legal literacy to promote positive social norms related to HIV, and routinely assessing knowledge, attitudes and practices of health-care workers towards vulnerable populations to support health facility administrators to identify and address any issues [61].

Lower levels of treatment optimism were associated with never having tested for HIV. One study reported that increased "treatment optimism" was associated with having condomless sex [62] and in this study we found an association between condomless sex in the past 3 months and HIV testing, with the likelihood of testing being higher among those who reported condom less sex. There is need to investigate further how "treatment optimism" affects HIV testing.

## Study limitations

We acknowledge our study has some limitations. Firstly, since information on HIV testing, stigma, health status among other variables is self-reported there is a possibility of social desirability bias. We endeavoured to minimise this by using ACASI. Secondly, data was collected from only 6 rural districts, and this might limit generalizability of findings to other districts especially non rural populations. Also, due to the cross-sectional nature of the study, causality between the outcome and some independent variables is difficult to assess. Another limitation is the exclusion of (74/11,150) 0.66% participants with inconsistency responses. However, the proportion is relatively small such that it is unlikely to have an impact on the validity of the results. Lastly, data analysed was collected in 2018-19, thus the prevalence of never testing has likely decreased in the meantime. This study focusses on the general population, however it would be important to understand whether this is similar among key populations which are an important group contributing to the HIV testing gap.

## Conclusion

We found that never testers were more likely to be males, of age below 35 or above 45, with low level of education, unemployed, never married, those projecting high level of stigma, lower level of HIV treatment optimism, those who were not participating in decision making. As the country aims to meet the 2030 UNAIDS global targets, MOHCC need to scale up

acceptable and targeted interventions to improve HIV testing in different subpopulations. The interventions include scaling up distribution of information, education and communication materials on HIV related topics in schools and communities, in local languages and through social media and community campaigns to reduce stigma associated with testing for HIV and targeting men and young people through initiatives such as men's clinics and workplace programs. Provision of knowledge on where HIV testing services can be found through media and mobile HIV testing campaigns has been reported to increase uptake of testing among men and young people especially in rural areas [63]. It is also crucial to increase HIVST delivery models both at facilities and communities, as this enhances testing uptake by offering more choices to clients [34].

## Supporting information

**S1 Table. Explanatory variables and their levels.**
(DOCX)

**S1 Text. Inclusivity in Global Research Checklist.**
(DOCX)

**S2 Table. Derivation of construct variables.**
(DOCX)

**S3 Table. Risk factors associated with never testing for HIV.**
(DOCX)

**S4 Table. Descriptive characteristics of those reporting no perception of HIV risk.**
(DOCX)

**S1 Data. Data set.**
(DTA)

**S1 File. Data dictionary.**
(PDF)

## Acknowledgments

We acknowledge the study participants, community leaders, study staff and Ministry of Health and Child Care.

## Author contributions

**Conceptualization:** Wellington Murenjekwa, Frances M Cowan, Cheryl Johnson, Amon Mpofu, Getrude Ncube, Owen Mugurungi, Karin Hatzold, Elizabeth L Corbett, Euphemia Sibanda, Valentina Cambiano.

**Formal analysis:** Wellington Murenjekwa, Fiona C Lampe, Andrew N Phillips, Valentina Cambiano.

**Funding acquisition:** Frances M Cowan, Karin Hatzold, Elizabeth L Corbett, Valentina Cambiano.

**Methodology:** Wellington Murenjekwa, Frances M Cowan, Karin Hatzold, Elizabeth L Corbett, Euphemia Sibanda.

**Project administration:** Kudzai Chidhanguro, Euphemia Sibanda, Valentina Cambiano.

**Software:** Wellington Murenjekwa.

**Supervision:** Frances M Cowan, Fiona C Lampe, Andrew N Phillips, Euphemia Sibanda, Valentina Cambiano.

**Visualization:** Wellington Murenjekwa.

**Writing – original draft:** Wellington Murenjekwa.

**Writing – review & editing:** Kudzai Chidhanguro, Frances M Cowan, Fiona C Lampe, Cheryl Johnson, Amon Mpofu, Getrude Ncube, Owen Mugurungi, Karin Hatzold, Elizabeth L Corbett, Andrew N Phillips, Euphemia Sibanda, Valentina Cambiano.

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
