## [Decision Letter · Decision Letter 0]

13 Nov 2024

PGPH-D-24-01522

“Who has never tested for HIV? Associated factors among people living in rural areas of Zimbabwe after community distribution of HIV self-testing kits”.

Dear Dr. Murenjekwa,

Thank you for submitting your manuscript to PLOS Global Public Health. After careful consideration, we feel that it has merit but does not fully meet PLOS Global Public Health’s publication criteria as it currently stands. Therefore, we invite you to submit a revised version of the manuscript that addresses the points raised during the review process.

We look forward to receiving your revised manuscript.

Kind regards,

Tsitsi B. Masvawure, Ph.D.

Academic Editor

Journal Requirements:

 1. Please include a complete copy of PLOS’ questionnaire on inclusivity in global research in your revised manuscript. Our policy for research in this area aims to improve transparency in the reporting of research performed outside of researchers’ own country or community. The policy applies to researchers who have travelled to a different country to conduct research, research with Indigenous populations or their lands, and research on cultural artefacts. The questionnaire can also be requested at the journal’s discretion for any other submissions, even if these conditions are not met.  Please find more information on the policy and a link to download a blank copy of the questionnaire here: https://journals.plos.org/globalpublichealth/s/best-practices-in-research-reporting. Please upload a completed version of your questionnaire as Supporting Information when you resubmit your manuscript. 2. In the online submission form, you indicated that "Data will be available upon request".  All PLOS journals now require all data underlying the findings described in their manuscript to be freely available to other researchers, either 1. In a public repository, 2. Within the manuscript itself, or 3. Uploaded as supplementary information. This policy applies to all data except where public deposition would breach compliance with the protocol approved by your research ethics board. If your data cannot be made publicly available for ethical or legal reasons (e.g., public availability would compromise patient privacy), please explain your reasons by return email and your exemption request will be escalated to the editor for approval. Your exemption request will be handled independently and will not hold up the peer review process, but will need to be resolved should your manuscript be accepted for publication. One of the Editorial team will then be in touch if there are any issues. 3. Please provide an Author Summary. This should appear in your manuscript between the Abstract (if applicable) and the Introduction, and should be 150–200 words long. The aim should be to make your findings accessible to a wide audience that includes both scientists and non-scientists. Sample summaries can be found on our website under Submission Guidelines:  https://journals.plos.org/globalpublichealth/s/submission-guidelines#loc-parts-of-a-submission 4. We have noticed that you have uploaded Supporting Information files, but you have not included a list of legends. Please add a full list of legends for your Supporting Information files after the references list.  

Additional Editor Comments (if provided):

Dear Authors,

Thank you so much for your patience. We have now successfully secured two reviews for your paper. As you can see, one reviewer focused primarily on your statistical methods and raises several concerns and suggestions. I invite you to respond to these in your revision. The other reviewer commented mostly on the broader implications of your study and they would like you to consider the "so what" of your findings. I would like to add that it may help if you describe the community intervention early on in the introduction or methods section, so that it is clear that data was collected from communities that had received the community HIV self-testing intervention. It took me some time to figure this out. You may also need to explain in a bit more detail how the HIV self-testing intervention was marketed and how community members were made aware of this initiative.

Sincerely,

Academic Editor

Reviewers' comments:

Reviewer's Responses to Questions

**Comments to the Author**

1. Does this manuscript meet PLOS Global Public Health’s publication criteria ? Is the manuscript technically sound, and do the data support the conclusions? The manuscript must describe methodologically and ethically rigorous research with conclusions that are appropriately drawn based on the data presented.

Reviewer #1: Yes

Reviewer #2: Yes

2. Has the statistical analysis been performed appropriately and rigorously?

Reviewer #1: Yes

Reviewer #2: Yes

3. Have the authors made all data underlying the findings in their manuscript fully available (please refer to the Data Availability Statement at the start of the manuscript PDF file)?

Reviewer #1: No

Reviewer #2: No

4. Is the manuscript presented in an intelligible fashion and written in standard English?

Reviewer #1: Yes

Reviewer #2: Yes

5. Review Comments to the Author

Reviewer #1: General comments

The manuscript is well written, an interesting read and provides a unique insight into an important topic in an exciting field of HIV self-testing delivery models in general populations in Zimbabwe. In their analysis, authors clearly identify sub-groups within the general populations who did not test for HIV despite the wider availability of HIV self-test kits in the community. A major observation is that there is limited discussion on implications of scaled distribution of HIV self-test kits in the community on the number of those who have never tested. Again, authors are not clear on specific combinations of attributes of sub-populations that are important to implementers who are intending to provide targeted distribution of HIV self-test kits in the community.

Specific comments

Title

Title needs rephrasing: 'Who has never tested for HIV following a community-based distribution of HIV self-test kits? Establishing associated predictors in rural Zimbabwe

Abstract

Line 33: Indicate the year when this was first achieved

Line 34: Rephrase: 'some groups of people...'

Line 35: Rephrase to 'we estimated the prevalence and predictors of community members who with a 'never-having tested for HIV' status following community-based distribution of HIV self-test kits in rural Zimbabwe.

Introduction

Line 64-65: The statement requires a reference

Lines 65-67: Rephrase the statement 'Globally, the burden of HIV is highest within the Eastern and Southern Africa (ESA) region which accounts for more than half (54%) of all people living with HIV (PLHIV) in the world [1]; but the region only has 7% of the global population[2].

Lines 68-68: Please rephrase: ' Although a considerable decline in HIV incidence (-44%) and AIDS related deaths (-58%) were estimated between 2010 and 2021 in ESA compared with other ...'

Lines 71-73: Kindly provide reasons why there has been such a dramatic decline in HIV incidence and prevalence in Zimbabwe.

Lines 77-78: Kindly describe these inequalities and why they exist?

Lines 79-83. I suggest breaking the sentence into two parts. One part should be about increased access to HIV testing services and the other part should be about HIV self-testing as an additional option.

Line 86: Add Zimbabwe to the sentence. i.e. 'The annual number of conventional HIV test performed in Zimbabwe...'

Line 88-89: It would be great to have more current estimates that are consistent with the first 95% that Zimbabwe has achieved.

Lines 91-92: Is HIVST part of the conventional HIV testing as described in line 86-87?

Lines 91-92: Is HIVST part of the conventional HIV testing as described in line 86-87?

Line 93: There is a need for a comma between the word 'test' and 'there'

Lines 96-97: Kindly use past tense i.e. 'This study estimated the prevalence of never having tested for HIV...'

Lines 98-99: 'We also assessed factors associated with reporting no

99 perceived HIV risk among participants who never tested for HIV.'

Methods

Line 103: Rephrase the sentence 'The study analysed data from a cluster randomised trial...'

Line 122: ...did not share a health facility with a neighboring cluster...

Lines 125-128: The sentence is incoherent. Please review and revise it

Line 129: Kindly use the word participants' instead of the word ‘participant’.

Line 162: Use the word 'made' instead of the word 'make'

Lines 176-177: Something is missing from this sentence. Kindly review and rephrase it.

Line 188: The use of the word range, there is an expectation to have two values i.e. the lowest value and the highest value.

Discussion

Lines 278-279: Please state 'in Zimbabwe' at the end of the sentence. I also think that merging the first and second sentence would read better. i.e. ' We found that 17% of the participants had never had an HIV test, in rural communities in Zimbabwe between 2018-2019 despite a community-based distribution of HIV self-testing kits which greatly improved access to HIV testing'

Line 280: Kindly delete the word 'however' from the sentence. Also replace the word 'the' at the beginning of the sentence with the word 'these results'

Line 286: These references should be merged i.e. [23-27]

Line 287: This reference is about men in West Africa. Kindly use a reference from the East and Southern Africa region. J K Chikovore has written a bit on masculinities and health seeking behaviout in this region

Lines 293-303. Results shows that 16% of 45+ also never tested. The authors have not discussed this component in this paragraph.

Line 315-316: I suggest leaving the reasons why unemployed were unlikely to self-test as a research gap that needs qualitative research to investigate.

Lines 322-323: Can you support this statement with references on the relationship between perceived low risk of contracting HIV and non-uptake of HIV testing.

Lines 330-332: The International AIDS Society has done some fantastic work in the area of HIV stigma 'Getting to the Heart of Stigma'. They do have great references on how to address stigma in the context of HIV. Kindly explore if you can use some of their references. See: https://www.iasociety.org/sites/default/files/JIAS/JIAS_Vol25-S1_complete_file.pdf

References

Line 482-483: Reference is quite old. Kindly use more recent papers

Reviewer #2: Overall, this work appears to highlight key demographic variables that are predictive of whether an individual tests for HIV. However more detail is needed to explain and justify statistical methodology and further interpretation of key results including reporting point estimates/p-values/confidence intervals in the main text would be helpful.

Authors mention the frequency of missing data was less than 1.5% for all variables, except one with 5% missing. Overall, authors should report the total percent of missing observations that were excluded from analyses. And were these observations excluded only in the analyses for which the missing variable was included in the model? Or were observations with any missingness dropped entirely from all analysis? This needs to be clarified.

For the principal component analysis, it would be helpful to report the percent of the variance that is explained by the first principal component which is included in analyses. This allows us to know how much information is lost by using the single summary measure rather than the individual items.

It is unclear why the authors choose to create quantile cutoffs and categorize the resulting scores from the PCA analyses for the wealth, participation in decisions, community cohesion, stigma, treatment optimism explanatory variables. Why not use the continuous scores obtained from PCA? If this decision was done to reflect previous literature, it should be stated.

Overall, the section labeled ‘Construct explanatory variables’ was confusing to read. I would recommend first outlining the procedure (PCA, then division of scores into categories) that was used across all the constructed variables, and then provide details on the summarization of each distinct measure.

On line 206 need to state how mixed effect model was specified (mixed effects by cluster? Random intercept only?)

Authors should explain why the two adjustment sets (sex and age vs sex, age, religion, education) were selected and what they expect either model to show.

Some more discussion might be given to the point on lines 220-222. It seems this might impact the overall validity of results, as it would seem there is some measurement error in the outcome. Is there any reason why this might be happening? This should be mentioned in the limitations in the discussion.

In Table 1, how are the p-values obtained? Does this analysis account for the correlation by cluster that is present in the data? If not, these results are not valid and should not be reported. If it does, this analysis should be equivalent to the unadjusted mixed effects model and would be repetitive to report in both tables.

For all models, the choice of reference groups is confusing and inconsistent. For example, selecting age 34-44 as the reference group and comparing to those older and younger is unusual, as typically the reference is selected as the most extreme category (either youngest or oldest, in this example). For the constructed variables, it would make sense to select consistently across all either low or high as the reference. The selection of reference groups here feels a bit like the authors might be arbitrarily searching for significant results.

Rather than just report the significance in the main text of the results, it would be helpful to also report p-values and or the effect estimates with confidence intervals to highlight important findings.

Smaller concerns

Weird/unclear phrasing examples

o Line 146 “Also, below is a summary of how certain variables were derived.”

o Line 156-157 “and took scores for the first principal component as household scores”

o Line 354, interpretation should be: males, age below above … were more likely to be never testers

Would recommend using quotes around variable names

o For example, on line 160, write ‘participation in decisions’

6. PLOS authors have the option to publish the peer review history of their article (what does this mean? ). If published, this will include your full peer review and any attached files.

**Do you want your identity to be public for this peer review?** For information about this choice, including consent withdrawal, please see our Privacy Policy .

Reviewer #1: No

Reviewer #2: No

---

## [Decision Letter · Decision Letter 1]

12 Mar 2025

"Who has never tested for HIV following a community-based distribution of HIV self-test kits? Establishing associated predictors in rural Zimbabwe."

PGPH-D-24-01522R1

Dear Mr Murenjekwa,

We are pleased to inform you that your manuscript '"Who has never tested for HIV following a community-based distribution of HIV self-test kits? Establishing associated predictors in rural Zimbabwe."' has been provisionally accepted for publication in PLOS Global Public Health.

Best regards,

Julia Robinson

Executive Editor

Reviewer Comments (if any, and for reference):

Reviewer's Responses to Questions

**Comments to the Author**

1. If the authors have adequately addressed your comments raised in a previous round of review and you feel that this manuscript is now acceptable for publication, you may indicate that here to bypass the “Comments to the Author” section, enter your conflict of interest statement in the “Confidential to Editor” section, and submit your "Accept" recommendation.

Reviewer #1: All comments have been addressed

Reviewer #2: All comments have been addressed

2. Does this manuscript meet PLOS Global Public Health’s publication criteria ? Is the manuscript technically sound, and do the data support the conclusions? The manuscript must describe methodologically and ethically rigorous research with conclusions that are appropriately drawn based on the data presented.

Reviewer #1: Yes

Reviewer #2: Yes

3. Has the statistical analysis been performed appropriately and rigorously?

Reviewer #1: I don't know

Reviewer #2: Yes

4. Have the authors made all data underlying the findings in their manuscript fully available (please refer to the Data Availability Statement at the start of the manuscript PDF file)?

Reviewer #1: Yes

Reviewer #2: Yes

5. Is the manuscript presented in an intelligible fashion and written in standard English?

Reviewer #1: Yes

Reviewer #2: Yes

6. Review Comments to the Author

Reviewer #1: I am happy with the changes made. The authors can just proofread the manuscript.

Reviewer #2: The authors made sufficient changes to address my original concerns.

7. PLOS authors have the option to publish the peer review history of their article (what does this mean? ). If published, this will include your full peer review and any attached files.

**Do you want your identity to be public for this peer review?** For information about this choice, including consent withdrawal, please see our Privacy Policy .

Reviewer #1: No

Reviewer #2: No
